# Gut microbiome dysbiosis drives metabolic dysfunction in Familial dysautonomia

Alexandra M. Cheney [1,4], Stephanann M. Costello[1,4], Nicholas V. Pinkham[2,4], Annie Waldum[1], Susan C. Broadaway[2], Maria Cotrina-Vidal[3], Marc Mergy[2], Brian Tripet[1], Douglas J. Kominsky[2], Heather M. Grifka-Walk[2], Horacio Kaufmann [3], Lucy Norcliffe-Kaufmann[3], Jesse T. Peach [1], Brian Bothner[1], Frances Lefcort[2] ✉, Valérie Copié [1] ✉ & Seth T. Walk [2] ✉

Familial dysautonomia (FD) is a rare genetic neurologic disorder caused by impaired neuronal development and progressive degeneration of both the peripheral and central nervous systems. FD is monogenic, with >99.4% of patients sharing an identical point mutation in the elongator acetyltransferase complex subunit 1 (*ELP1*) gene, providing a relatively simple genetic background in which to identify modifiable factors that influence pathology. Gastrointestinal symptoms and metabolic deficits are common among FD patients, which supports the hypothesis that the gut microbiome and metabolome are altered and dysfunctional compared to healthy individuals. Here we show significant differences in gut microbiome composition (16 S rRNA gene sequencing of stool samples) and NMR-based stool and serum metabolomes between a cohort of FD patients (~14% of patients worldwide) and their cohabitating, healthy relatives. We show that key observations in human subjects are recapitulated in a neuron-specific *Elp1*-deficient mouse model, and that cohousing mutant and littermate control mice ameliorates gut microbiome dysbiosis, improves deficits in gut transit, and reduces disease severity. Our results provide evidence that neurologic deficits in FD alter the structure and function of the gut microbiome, which shifts overall host metabolism to perpetuate further neurodegeneration.

From 1791 to 1917, Jews living in the Russian Empire were forced to live inside the prescribed Pale of Settlement, a region spanning present day Ukraine, Poland, Slovakia, Hungary, Republic of Moldova, and Romania. This oppression created generations of endogamy within the region and a genetic bottleneck that enriched a deleterious allele of the *ELP1* gene such that today, between 1 in 18 to 1 in 27 Ashkenazi Jews carry the original *ELP1* founder mutation (c2204 + 6 T > C)[1–3]. The allele is highly penetrant and results in impaired splicing of *ELP1* (formerly called *IKBKAP*[4,5]), leading to a devastating neurologic disease first described in 1949[6] and now called Familial dysautonomia (FD). FD

patients are homozygous for the mutant allele and have tissue-specific reductions in Elp1 protein, primarily in the nervous system[7,8]. Elp1 is a key scaffolding subunit of the six-protein Elongator complex and is required for codon-biased mRNA translation with additional reports of a role in transcript elongation[9–13]. FD symptoms result largely from dysfunctional neuronal development and neurodegeneration of sensory and autonomic neurons in the peripheral nervous system, leading to widespread deficits in organ innervation[3]. FD patients have reduced body mass index (BMI), high metabolic rate, and low fat content indicative of metabolic deficits[14]. In addition, patients have severe

[1]Department of Chemistry and Biochemistry, Montana State University, Bozeman, MT, USA. [2]Department of Microbiology and Cell Biology, Montana State University, Bozeman, MT, USA. [3]Department of Neurology, New York University School of Medicine, New York, NY, USA. [4]These authors contributed equally: Alexandra M. Cheney, Stephanann M. Costello, Nicholas V. Pinkham. ✉e-mail: lefcort@montana.edu; vcopie@montana.edu; seth.walk@montana.edu

cardiovascular, respiratory, and gastrointestinal (GI) dysfunction due to innervation deficits[3,15,16]. GI complaints are frequently reported by FD patients and these symptoms significantly impact their quality of life[17]. Similarly, patients with other neurological disorders, such as autism spectrum disorder (ASD), also suffer from debilitating GI symptoms and share hallmark neurological impairments with FD[18–20]. The ultimate therapeutic goal for FD is to develop strategies to increase levels of Elp1 (protein) in the affected cell populations and tissues, but no such treatments are clinically available. Since the peripheral nervous system senses and responds to both the gut microbiome and liver – and in response to these signals regulates digestion and metabolism, we sought to answer the basic question: Is the gut-metabolism axis involved in FD pathology?

To address this question, we recruited FD patients and healthy, cohabitating relatives to help control for lifestyle factors known to influence comparisons[21], such as environment (cohabitants share a built environment), diet (cohabitants are more likely to eat the same food), and genetics (parents carry one copy of the deleterious *ELP1* allele and siblings have a 50% chance of carrying the mutant allele). In total, 49 patients and 54 relatives provided at least one sample for microbiome (*n* = 51 patient-relative combinations) or metabolome analyses (n = 58 patient-relative combinations for stool; n = 50 patient-relative combinations for serum; Supplementary Table 1a, b in Supplementary Data 1). Given an estimated 350 FD patients worldwide, this sampling effort represents ~14% of all known FD patients and a sample size comparable to those reported in previous gut microbiome studies of autism spectrum disorder (ASD), Alzheimer's disease, and Parkinson's disease[22–25]. Here, we present evidence that the gut-metabolism axis is an important factor in FD progression and can be at least partially reversed by microbiome homogenization via co-housing in a mouse model of the disease.

## Results

### Altered gut-metabolism axis in FD

Gut microbiome diversity was quantified from stool samples using Illumina sequencing of the bacterial 16 S rRNA encoding gene (V4 region) and polar metabolites (serum and stool) were identified and quantified using [1]H NMR-based metabolomics. Results from such '-omics' technologies are typically considered with respect to diversity, meaning the presence-absence and relative abundance of observations (taxa for microbiome sequencing and metabolites for metabolomics). Both within the sample (alpha-) and between group (beta-) diversity estimates are important. All microbiome and metabolome samples were plotted using ordination to visually compare groups (Fig. 1a, e, f). Statistical tests were then performed to evaluate the significance and identify key bacterial operational taxonomic units (OTUs; ≥97% 16 S rRNA gene sequence identity) or specific metabolites that explained observed differences between FD patients and relatives. Clear groupwise differences were apparent in all ordinations, which were supported by PERMANOVA testing. FD patient microbiomes were more variable (greater dispersion, Fig. 1b; distance to group centroid, Supplementary Fig. 1b) and represented only a subset of species found in paired relatives (i.e., FD patients shared more OTUs with relatives than their relatives shared with them, and FD patients had an average of 45 fewer bacterial taxa compared to paired relatives; Fig. 1d). Alpha-diversity was also lower in FD patient microbiomes compared to paired relatives (Fig. 1a inset and Supplementary Fig. 1a, c). We observed a positive trend between alpha diversity of FD patients and their relatives (size of connected dots in Fig. 1a), and while intriguing, this did not reach statistical significance (linear mixed effects model, Chi-square = 3.110, df = 1, *p* = 0.078). Partition around medoids (PAM) clustering of OTU-based diversity suggested that microbiomes from all subjects (i.e., patients and relatives) represented two distinct community types, or enterotypes (Supplementary Fig. 2a) with FD patients significantly associated with cluster 1 and relatives with cluster 2

(Fisher's exact test, Odds ratio = 9.388, 95% confidence interval of odds ratio = 2.444 to 54.000); Supplementary Fig. 2b). The cluster 1 enterotype exhibited significantly lower alpha diversity compared to cluster 2 (inverse-Simpson's index, t-test, t-statistic = −7.028, df = 67.78, *p* < 0.0001, mean difference between patients and relatives = −7.884, 95% confidence interval of the difference between patients and relatives = −10.123 to −5.646). Also, the abundance of seven OTUs

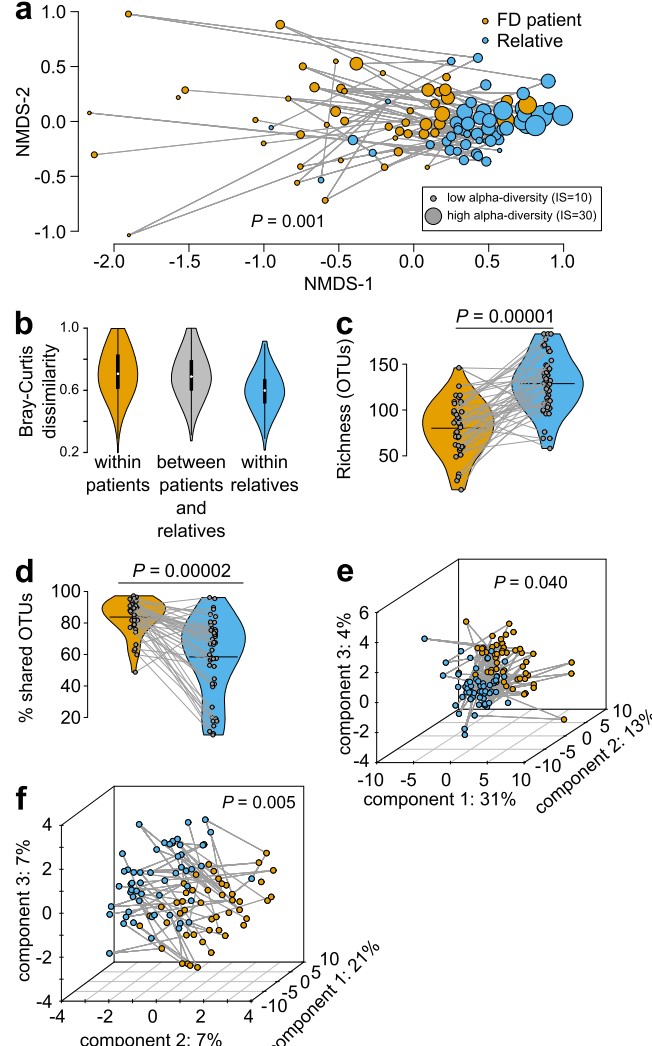

**Fig. 1 | FD alters the microbiome and metabolome. a** Non-metric multidimensional scaling (NMDS) of 16 S rRNA-based stool microbiome diversity in FD patients and healthy relatives (all patient-relative pairs connected with lines). IS, inverse-Simpson's index. **b**, Microbiome beta-diversity (Bray-Curtis dissimilarity) within and between FD patients and paired relatives (each violin plot contains a box and whisker plot here a white dot = median, box = interquartile range (IQR), whiskers = 1.5 times IQR). **c**, **d**, Microbiome richness (**c**) and percentage (%) of shared operational taxonomic units (OTUs) (**d**) between FD patient and healthy relatives (colors as in **a**; bars = mean; lines connect patients and relatives). **e**, **f** Partial least-squares discriminant analysis (PLSDA) of [1]H-NMR-based stool (**e**) and serum (**f**) metabolome diversity in FD patients and healthy relatives (colors as in **a**, **b**). *n* = **a**, **b**, 48 patients, 51 relatives; **c**, **d**, 38 patients, 47 relatives, 47 pairs; **e**, 54 patient, 58 relatives; **f**, 49 patients, 53 relatives. PERMANOVA of Bray-Curtis dissimilarity (**a**, F = 6.847, df = 1, R² = 0.066) or Euclidean distance (**e**, F = 2.153, df = 1, R² = 0.0191; **f**, F = 2.506, df = 1, R² = 0.0245); permutational paired t-testing, two-sided (**c**, mean t-statistic = −5.423, df = 29, mean difference between group means = −45.114, mean 95% confidence interval of the difference between the group means = −62.130 to −28.100; **d**, mean t-statistic = 5.220, df = 29, mean difference between group means = 24.944, mean 95% confidence interval of the difference between the group means = 15.16 to 34.73).

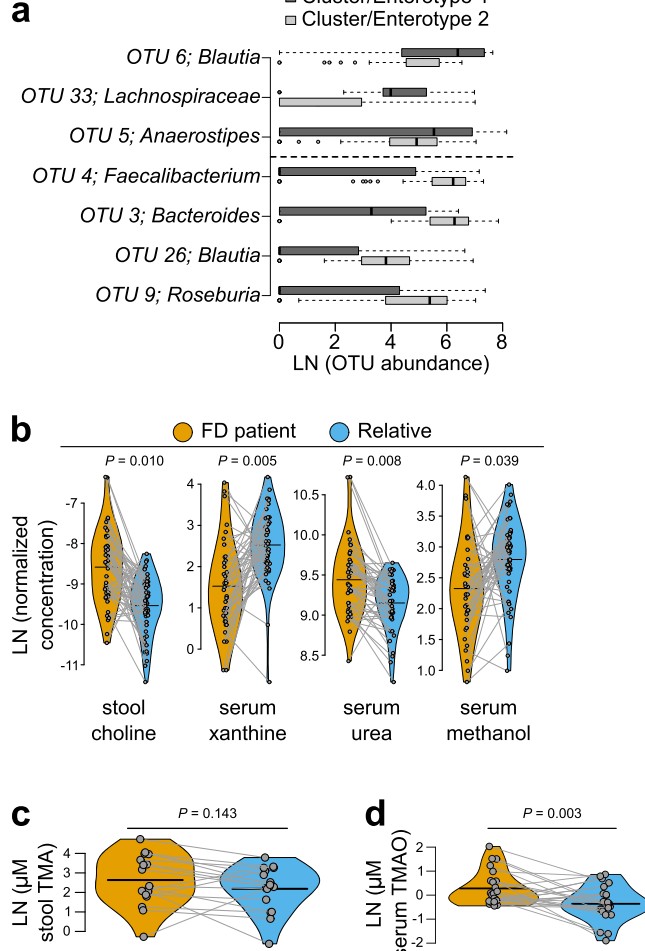

**Fig. 2 | Microbiome OTU and metabolite differences in FD. a** OTUs responsible for observed partition around medoids (PAM) clusters/enterotypes identified by random forest analysis. OTUs defining cluster/enterotype 1 are shown above the dotted line. OTUs defining cluster/enterotype 2 are shown below the dotted line (centrality line = median, boxes = IQR, whiskers = 1.5 times IQR, dots beyond whiskers = outliers). **b** Metabolites showing significantly different abundances in FD patients compared to relatives (line = mean). **c** Targeted analyses of TMA in stool (line = mean). **d** Targeted analysis of TMAO in serum (line = mean). n = **a**, 48 patients, 51 relatives; **b**, stool, 54 patient, 58 relatives; serum, 49 patients, 53 relatives; **c**, **d**, 24 patient-relative pairs. Random forest analysis (**a**); Natural log (LN) transformed metabolite concentration, permutational paired t-testing (two-sided) with FDR correction (**b**, choline, mean t-statistic = 0.1485, df = 33, mean difference between patient-relative pairs = 0.9860, mean 95% confidence interval of difference between patient-relative pairs = 0.5281 to 1.4440; xanthine, mean t-statistic = 2.9527, df = 33, mean difference between patient-relative pairs = −0.9591, mean 95% confidence interval of difference between patient-relative pairs = −1.4051 to −0.5130; urea, mean t-statistic = 0.9269, df = 33, mean difference between patient-relative pairs = 0.7281, mean 95% confidence interval of difference between patient-relative pairs = 0.3527 to 1.1036; methanol, mean t-statistic = −1.5497, df = 33, mean difference between patient-relative pairs = −0.7295, mean 95% confidence interval of difference between patient-relative pairs = −1.1875 to −0.2716); Paired t-testing, two-sided) (**c**, TMA, t-statistic = 1.5423, df = 16, difference between patient-relative pairs = 0.4499, 95% confidence interval of difference between patient-relative pairs = −0.1685 to 1.0682; **d**, TMAO, t-statistic = 3.3197, df = 23, difference between patient-relative pairs = 0.6418, 95% confidence interval of difference between patient-relative pairs = 0.2419 to 1.0417).

identified by random forest analysis and typically associated with human health were altered, four of which were less abundant in cluster 1 (Fig. 2a and Supplementary Fig. 3) and among FD patients in general, including *Faecalibacterium*[26] (OTU 4; permutational paired t-test,

mean t-statistic = −2.247, df = 29, mean *p* = 0.0360, mean difference between patient-relative pairs = −220.51, mean 95% confidence interval of the difference between patient-relative pairs = −421.102 to −19.918) and *Roseburia*[27] (OTU 9; permutational paired t-test, mean t-statistic = −2.093, df = 29, mean p = 0.0496, mean difference between patient-relative pairs = −173.801, mean 95% confidence interval of the difference between patient-relative pairs = −343.562 to −4.040). Overall, these results suggest that many FD patients lacked common "healthy" microbiome members.

All relatives of FD patients that participated in this study as matched controls were known to carry one copy of the *ELP1* founder mutation (previously screened at the Dysautonomia Center, NYU Langone Health) and even though carriers are asymptomatic with respect to typical FD symptoms[3], there is a possibility that the founder allele is penetrant at the level of microbiome diversity. If true, microbiomes of FD relatives would be different from those of other healthy adults. To test this hypothesis, we evaluated 16 S rRNA sequencing data from healthy subjects of two previously published studies[28,29]. Subjects from the first study[28] (n = 172) were part of a large, multicenter, cross-sectional design and subjects from the second study[29] (n = 8) were sampled longitudinally for an average of 512 days (average of 49 samples per subject), thereby allowing us to consider both between- and within-subject microbiome diversity. When raw reads from these studies were combined and compared to FD relatives (Supplementary Fig. 4,a, c) and FD patients (Supplementary Fig. 4,b, c), beta-diversity (Bray-Curtis dissimilarity) was significantly different. However, when analyzed separately and compared, the microbiome diversity between all cohorts was significantly different (Supplementary Fig. 4,d) and based on the distance to group centroids (the weighted average microbiome diversity of each group, Supplementary Fig. 4,e), differences were greater between healthy individuals and FD patients compared to FD relatives, regardless of study. Collectively, these results provide little evidence that the gut microbiome diversity of FD relatives is significantly different from that of healthy subjects in other studies, where the frequency of the *ELP1* founder mutation is presumably low. Moreover, based on the observed distances to group centroids, the microbiome of FD patients is the most different of all cohorts analyzed, supporting a significant deviation from the microbiome diversity typically observed in healthy subjects.

Similar to microbiome diversity, metabolite levels were consistently different between patients and relatives (Fig. 1e, f). Of the 55 polar serum metabolites identified, FD patients had significantly lower levels of xanthine and methanol, and elevated levels of urea (Fig. 2b, Supplementary Fig. 5,a). Of the 73 polar stool metabolites identified, the only significant difference was an elevated level of choline in FD patients (Fig. 2b, Supplementary Fig. 5,b). Among FD patients, increased choline levels in stool were significantly correlated with lower microbiome alpha diversity, decreased richness, and fewer shared OTUs among patient-relative pairs (Supplementary Table 2 in Supplementary Data 1). Dietary choline is a precursor for microbiome-driven production of trimethylamine[30], which can be converted in the liver to the well-known risk factor for cardiovascular[31] and renal[32,33] disease, trimethylamine N-oxide (TMAO). To investigate whether elevated choline correlated with increased TMAO levels, we used a targeted liquid-chromatography mass-spectrometry (LCMS) method on a sub-set of patient-relative pairs (*n* = 24) and found that serum TMAO levels were indeed significantly elevated in FD patients (Fig. 2d), while TMA in stool trended in the same direction but did not reach statistical significance (Fig. 2c).

### Patient factors associated with dysbiosis

Important factors specific to FD and/or FD treatment likely contributed to differences in microbiome and metabolome diversity between FD patients and relatives and especially to the heterogeneity observed among FD patient samples. As might be expected of a

progressive neurodegenerative disease, patient age was statistically associated with several microbiome attributes (Supplementary Table 2 in Supplementary Data 1). In fact, of all factors evaluated in univariate analyses, patient age was the strongest correlate of microbiome diversity. Positive relationships were observed between both age and overall FD patient microbiome variability (dispersion) and similarity to the microbiome of relatives (beta-diversity between patient-relative pairs). A negative relationship was observed between age and the number of OTUs shared with relatives.

As FD progresses, most patients become unable to ingest food orally due to problems swallowing and a gastrostomy tube (G-tube) is placed as standard of care therapy. This procedure did have a significant effect (Supplementary Table 2 in Supplementary Data 1) on the level of choline in stool as FD patients with a G-tube ($n = 42$) had ~2-fold increased levels of choline compared to patients without a G-tube ($n = 8$), although the sample size of the latter group was low. Most FD patients continue to eat food orally for some time even after having a G-tube in place. As the inability to eat progresses, patients use a G-tube exclusively, and this factor (exclusive use of a G-tube) had a significant impact on gut microbiome diversity. Patients who used a G-tube exclusively had less diverse microbiomes compared to those able to consume food orally, shared fewer OTUs with relatives, and were 4.8 times as likely to have an FD-associated cluster 1 microbiome enterotype (Supplementary Table 2 in Supplementary Data 1). Given the progressive nature of FD, however, patient age and exclusive use of a G-tube are covariates and their independent effects are difficult to disentangle.

Antibiotic history also impacted microbiome diversity as patients taking antibiotics within three months prior to sampling were 4.4 times more likely to host an FD-associated cluster 1 microbiome compared to patients that were not treated with antibiotics (Supplementary Table 2 in Supplementary Data 1). Fundoplication, a gastric surgery to decrease acid reflux, was significantly associated with stool choline (Supplementary Table 2 in Supplementary Data 1). Sex, weight, and body mass index (BMI) were also evaluated as a potentially important factor but did not reach statistical significance for either microbiome or metabolome diversity. Finally, results from a clinical chemistry panel were available for some patients and levels of several analytes were associated with microbiome diversity (Supplementary Table 2 in Supplementary Data 1), including creatinine (positively correlated with beta diversity between patient-relative pairs; negatively correlated with the proportion of shared OTUs between patient-relative pairs); estimated glomerular filtration rate (positively correlated with richness; the number of OTUs shared with relatives; and the number of unique OTUs); and alanine aminotransferase (positively correlated with the proportion of shared OTUs between patient-relative pairs). As expected, blood urea nitrogen (BUN) was significantly associated with serum urea, serving as a type of internal control for NMR-based metabolomics results (Fig. 2b). Aspartate aminotransferase and alkaline phosphatase were also evaluated but not significantly associated. Collectively, these results suggest that both the progressive neurodegenerative effects of FD and the clinical history of patients are important determinants of gut-metabolism axis function.

### Neuronal Elp1 controls gut-metabolism axis

To more directly address the impact of Elp1-driven neurodegeneration on the gut-metabolism axis, we leveraged a previously described murine model of FD where *Elp1* is specifically ablated in approximately half of peripheral neurons (including enteric neurons) and in the vast majority of neurons of the central nervous system (*Tuba1a-cre⁺; Elp1^{loxp/loxp}* hereafter referred to as FD mice[34,35]). Compared to littermate controls (*Tuba1a-cre⁻; Elp1^{+/loxp}*), we found that the microbiome diversity of FD mice began diverging within 14 days post-weaning (DPW) when housed separately according to genotype (i.e., separately from control littermates; Fig. 3a, b). Similar to FD patients, the microbiome of FD

mice became more variable (greater dispersion) compared to controls by 54 DPW (Fig. 3c) and stool metabolite diversity, represented by 68 metabolites, was significantly divergent by 162 DPW (Fig. 3d). Collectively, results from these mice indicate that reduction of neuronal-Elp1 disrupts gut-metabolism homeostasis.

To address whether mutant *Elp1*-driven dysfunction can be rescued by controlling or restraining microbiome divergence, we compared FD mice in our colony that were reared in the presence or absence of littermate controls for different amounts of time (several cages per time point). This approach takes advantage of natural rodent coprophagy among cohoused animals known to homogenize microbiome diversity and recently shown to support normal metabolic functions, neurochemical homeostasis, and cognitive function[36]. These experiments were carried out for up to 329 days post-weaning so that signs of disease, if present, could be observed and quantified (early signs of disease in FD mice are typically observable around 3 months of age (~71 DPW). Similar to previous results (Fig. 3), the stool microbiome and metabolome of separately housed FD and control mice diverged in a statistically significant manner. In contrast, no differences were observed among cohoused mice at the same time point (79 DPW; Fig. 4a, b). Differences in microbiome diversity progressively increased over the next 200 days (increasing F-statistic) regardless of housing conditions but were far greater when the FD mice were housed separately from littermate controls (Fig. 4a). This divergence, even when cohoused, is noteworthy because it likely reflects the strength of the neuronal *Elp1* genetic deletion on the gut ecosystem. Differences in stool metabolite patterns between cohoused FD and control mice also increased with time, but never reached statistical significance throughout the experiment (Fig. 4b). Thus, it appears that cohousing did have a significant impact on the gut-metabolism axis in this murine model. Because FD mice, like FD patients, vary in their clinical phenotype[34,35], we developed a pathology scoring metric to quantify hallmark signs of the disease, such as abnormal/unhealthy body condition, hind limb clasping, and kyphosis (Supplementary Table 3) with sicker mice receiving higher numerical scores. Interestingly, cohousing significantly lowered pathology scores observed among FD mice (Fig. 4c; Supplementary Fig. 6). Moreover, cohousing ameliorated a deficit in FD mouse gut function : we found that gut transit time was significantly slower in separately housed FD mice compared to controls, but was significantly increased when FD and control mice were cohoused (Fig. 4d; Supplementary Fig. 7). Age (DPW) did not appear to contribute significantly to this effect (mouse age was not a significant factor when included in the analysis). Several metabolite level differences were observed between separately housed and cohoused FD mice that could account for differences in disease and gut pathology (Supplementary Fig. 8), including the observation that separately housed FD mice had elevated choline levels at the final time point (Day 279 DPW) compared to the same mice that were cohoused with control mice (Supplementary Fig. 8,a). We also observed elevated levels of the microbiome-derived choline metabolite, methylamine, at both 179 and 279 DPW (Supplementary Fig. 8,a), further supporting altered choline metabolism when FD mice were housed separately from controls. Since stool choline and TMAO were increased in human FD patients (Fig. 2b), this result suggests that despite the marked differences in diet and gut microbiome diversity between lab mice and humans, the impact of neuronal *Elp1* deletion on microbial choline metabolism may be conserved. Together, these results add experimental evidence that at least some of the neuronal *Elp1*-driven pathology and perturbation of the gut ecosystem in FD mice is modifiable by controlling microbiome-metabolome divergence.

## Discussion

Currently, the only treatment for FD is palliative care (pain and symptom management), and until therapies can be developed that

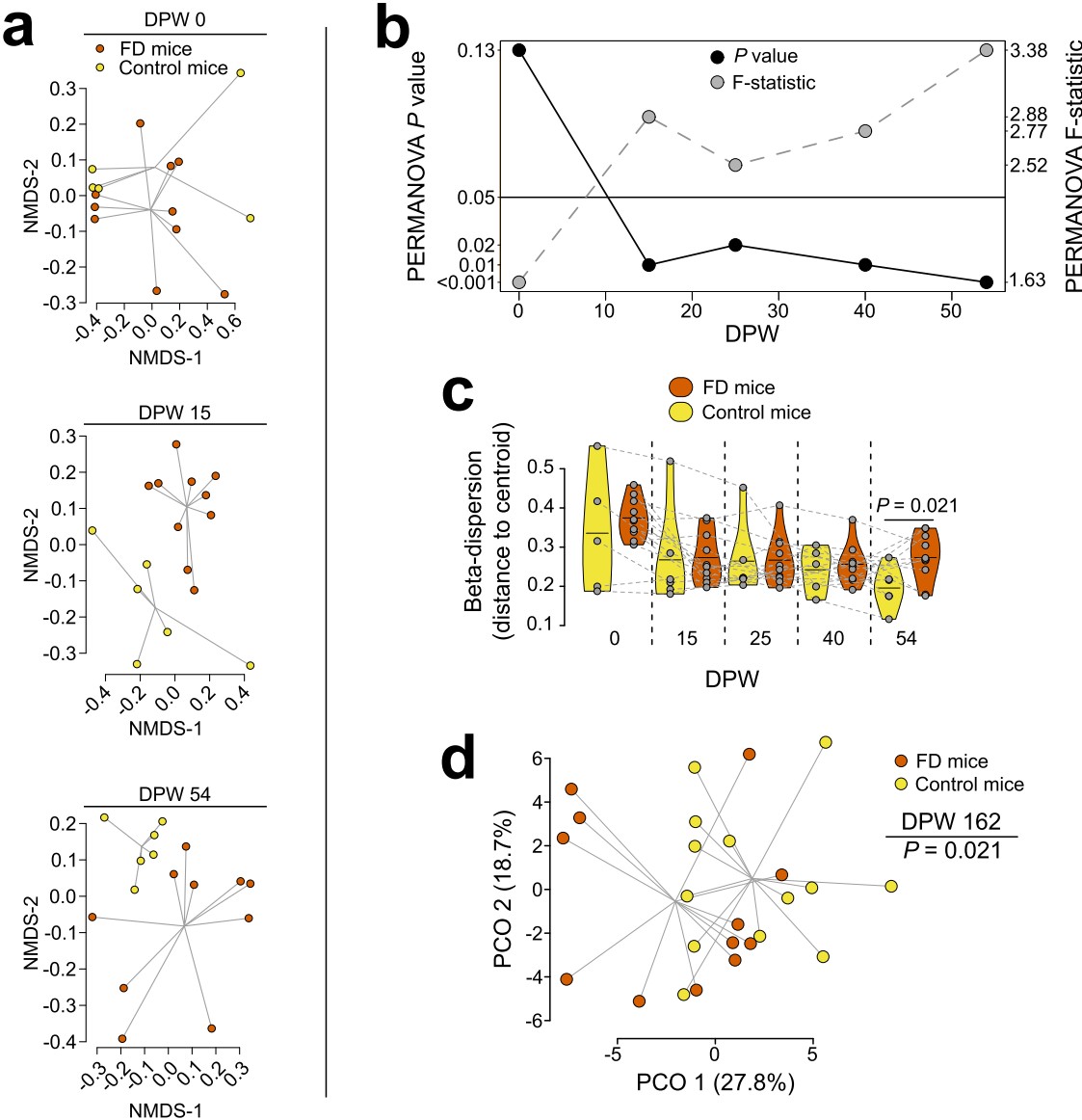

**Fig. 3 | Neuron-specific Elp1 deletion drives progressive microbiome and metabolome divergence in mice. a** Nonmetric multidimensional scaling (NMDS) of 16 S rRNA-based stool microbiome diversity of FD (*Tuba1a-cre⁺; Elp1^{loxp/loxp}*) and control (*Tuba1a-cre⁻; Elp1^{+/loxp}*) mice when housed separately by genotype (group colors same for all panels; DPW = days postweaning). **b** Progressive groupwise differences in microbiome beta-diversity (Bray-Curtis dissimilarity) and (**c**) dispersion of beta-diversity (beta-dispersion) in FD mice (dotted lines connect individual mice, solid line in violin plots = mean). **d** Principal coordinates analysis of stool metabolome in mice at 162 DPW. *n* = **a**, **b**, **c** 10 FD mice (4 female, 6 male; 3 cages), 6 control mice (4 female, 2 male; 2 cages); **d**, 12 FD mice, 13 control mice (all female; 3 cages per genotype); PERMANOVA (**a**, **b**, day 0, df = 1, R² = 0.1099; day 15, df = 1, R² = 0.1664; day 25, df = 1, R² = 0.1458; day 40, df = 1, R² = 0.1836; day 54, df = 1, R² = 0.1927; **d**, F = 2.2086, df = 1, R² = 0.0876); Welch's two-sample t-test (**c**, t-statistic = −2.6589, df = 12.05, difference between FD and control mice = −0.0777, 95% confidence interval of difference between FD and control mice = −0.1414 to −0.0141).

correct or replace the underlying genetic mutation and/or cellular dysfunction, identifying modifiable factors that slow pathology is paramount. Unlike other more common neurodegenerative diseases (Alzheimer's disease, Parkinson's disease, and amyotrophic lateral sclerosis, ALS), the genetic basis for FD is monogenic, highly penetrant, and recapitulated in neuron-specific mouse models. We have provided evidence that the FD-associated *ELP1* mutation promotes a dysfunctional gut-metabolism axis that in turn promotes pathology reminiscent of more common neurodegenerative diseases and other neurologic/neuropsychiatric conditions. Elp1 is an evolutionarily conserved protein among eukaryotes and a key component of the elongator complex that chemically modifies the wobble uridines in specific tRNA anticodon loops[10,37]. Loss of these modifications leads to translation inefficiency, altered proteome, and consequently cellular dysfunction and premature cell death.

Reduction of another elongator subunit, Elp3, also directly alters metabolism in yeast[38].

This is the first evaluation of the gut-metabolism axis in FD and the first of a disease involving degeneration of the peripheral nervous system. Several studies have investigated the contribution of the gut microbiome and/or metabolome to outcomes in other neurodegenerative diseases and at least three included patients' relatives as control subjects[39–41]. However, these previous studies were designed as cohort comparisons, and we are unaware of other microbiome or metabolomics studies in neurodegenerative diseases that compared patients with healthy relatives in a case-control design. As shown recently[21], host factors can confound gut microbiome studies and so matching or pairing is important to minimize spurious associations. Our results are consistent with clinical studies showing that virtually all FD patients have GI symptoms, including constipation, diarrhea and

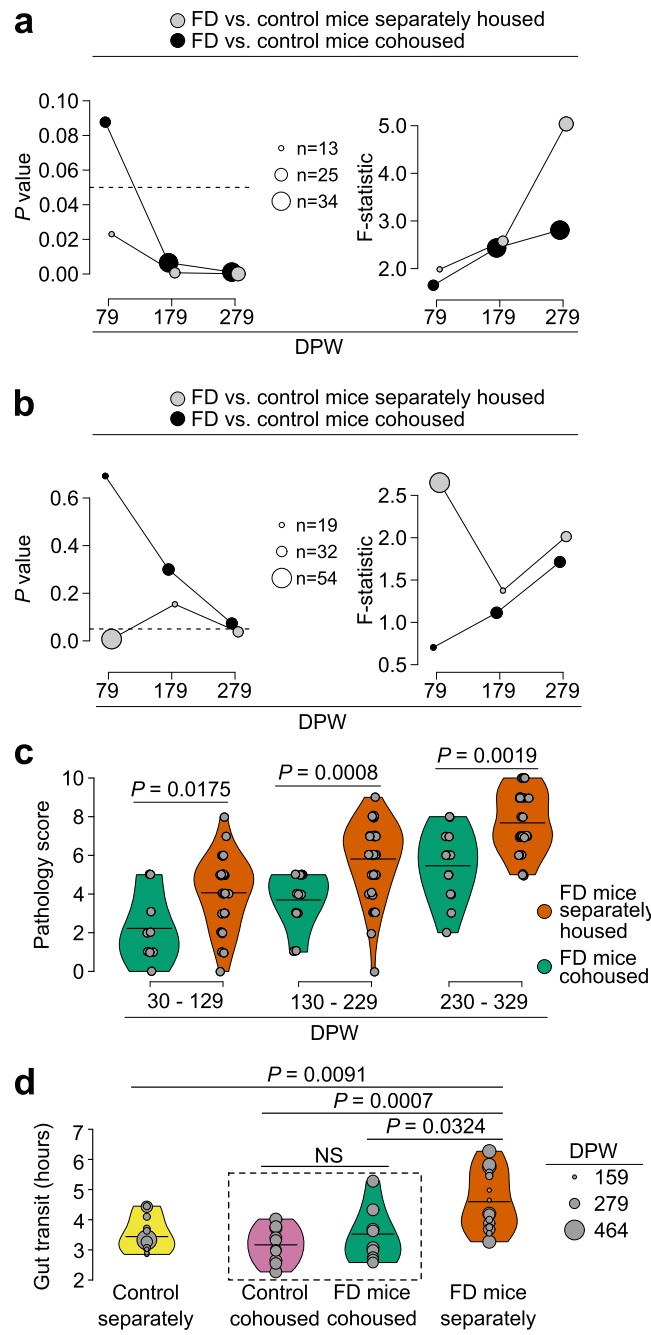

**Fig. 4 | Cohousing ameliorates gut transit deficit and disease in FD mice.**
**a**, **b** Progressive changes in microbiome (**a**) and metabolome (**b**) diversity in FD mice compared to control mice when separately housed or cohoused (size of circles = number of mice sampled at each time; DPW = days post-weaning; dotted line represents $p = 0.05$). **c** Pathology score of FD mice when housed separately or cohoused with control mice. **d** Carmine red-based gut transit for each mouse group. Groups in dotted box were cohoused. $n = $ **a**, 79 DPW separately housed: 10 FD (4 female, 6 male; 3 cages) vs. 3 control (female; 1 cage); 79 DPW cohoused: 9 FD (6 female, 3 male; 3 cages) vs. 12 control (7 female, 5 male; 3 cages); 179 DPW separately housed: 10 FD (5 female, 5 male; 2 cages) vs. 11 control (5 female, 6 male; 3 cages); 179 DPW cohoused: 13 FD (6 female, 7 male; 4 cages) vs. 21 control (7 female, 14 male; 5 cages); 279 DPW separately housed: 13 FD (6 female, 7 male; 4 cages) vs. 21 control (7 female, 14 male; 5 cages); 279 DPW cohoused: 13 FD (7 female, 6 male; 4 cages) vs. 8 control (5 female, 3 male; 2 cages). **b** 79 DPW separately housed: 25 FD (14 female, 11 male; 6 cages) vs. 29 control (18 female, 11 male; 7 cages) mice; 79 DPW cohoused: 9 FD (6 female, 3 male; 3 cages) vs. 11 control (6 female, 5 male; 3 cages); 179 DPW separately housed: 10 FD (6 female, 4 male; 3 cages) vs. 9 control (7 female, 2 male; 3 cages); 179 DPW cohoused: 13 FD (6 female, 7 male; 4 cages) vs. 21 control (7 female, 14 male; 5 cages); 279 DPW separately housed: 15 FD (9 female, 6 male; 4 cages) vs. 12 control (5 female, 7 male; 3 cages); 279 DPW cohoused: 13 FD (6 female, 7 male; 4 cages) vs. 20 control (7 female, 13 male; 5 cages). **c**, 30–129 DPW: 86 separately housed (60 female, 26 male; 18 cages) vs. 21 cohoused (13 female, 8 male; 5 cages); 130–229 DPW: 66 separately housed (47 female, 19 male; 14 cages) vs. 34 cohoused (13 female, 21 male; 8 cages); 230–329 DPW: 40 separately house (22 female, 18 male; 9 cages) vs. 34 cohoused (13 female, 21 male; 8 cages); **d**, 15 FD (12 female, 3 male; 3 cages) and 11 control (all female; 3 cages) mice separately housed; 9 FD (6 female, 3 male; 3 cages) and 10 control (7 female, 3 male; 3 cages) mice cohoused. PERMANOVA (**a**, separately housed, day 79, df = 1, $R^2$ = 0.1528, day 179, df = 1, $R^2$ = 0.1193, day 279, df = 1, $R^2$ = 0.1941; cohoused, day 79, df = 1, $R^2$ = 0.0799; day 179, df = 1, $R^2$ = 0.0706; day 279, df = 1, $R^2$ = 0.0807; **b**, separately housed, day 79, df = 1, $R^2$=, day 179, df = 1, $R^2$=, day 279, df = 1, $R^2$ = ; cohoused, day 79, df = 1, $R^2$=; day 179, df = 1, $R^2$=; day 279, df = 1, $R^2$=); Welch's two-sample t-testing with FDR correction (**c**, DPW 30–129, t-statistic = −2.7203, df = 13.018, difference between FD cohoused and separately housed mice = −1.8403, 95% confidence interval of difference between FD cohoused and separately housed mice = −3.3015 to −0.3790; DPW 130–229, t-statistic = −4.1497, df = 28.434, difference between FD cohoused and separately housed mice = −2.1185, 95% confidence interval of difference between FD cohoused and separately housed mice = −3.1635 to −1.0735; DPW 230–329, t-statistic = −3.7242, df = 20.515, difference between FD cohoused and separately housed mice = −2.2170, 95% confidence interval of difference between FD cohoused and separately housed mice = −3.4568 to −0.9772; **d**, FD cohoused vs. separately housed, t-statistic = −2.9745, df = 18.025, difference between group means = −1.1576, 95% confidence interval of difference between group means = −1.9751 to −0.3400; control cohoused vs. FD separately housed, t-statistic = −4.6345, df = 22.825, difference between group means = −1.4317, 95% confidence interval of difference between group means = −2.0710 to −0.7923; control separately housed vs. FD separately housed, t-statistic = −3.5146, df = 23.385, difference between group means = −1.0759, 95% confidence interval of difference between group means = −1.7087 to −0.4432).

gut dysmotility[17], and comparative pathology shows a reduced number of enteric neurons in FD patients as well as vagal nerve deficits in those with fundoplication[17]. Given the complex clinical histories of FD patients, including antibiotic treatments and GI procedures, it is reasonable to expect that the microbiome and metabolome would be significantly altered. Yet, neuron-specific deletion of *Elp1* in mice resulted in the progressive microbiome and metabolome alteration, suggesting that a neuronal FD mutation is sufficient to cause changes similar to that observed in human FD patients. It is also interesting that despite obvious differences between mice and humans (genetics, environment, diet, microbiome membership; Supplementary Fig. 9), choline metabolism was dysregulated in both human FD patients and the FD mouse model, suggesting a potentially important overlap in the mechanism of pathology.

While it is well known that FD patients suffer from frequent GI dysfunction[17], at least one imaging study found delayed esophageal transit times and delayed gastric emptying among FD patients compared to healthy control subjects, implicating a dysfunctional myenteric plexus of the upper GI tract in FD[42]. These results are consistent with two case studies that isolated and visually compared the esophageal, gastric and/or appendiceal myenteric plexus of FD patients and age-matched control subjects[43]; these studies showed significant reductions in the neuronal number and/or ganglion complexity. Collectively, these results suggest that dysfunctional gut motility in FD patients is most likely due to alterations in neuronal density throughout the GI tract. Like FD patients, our non-cohoused FD mice also had reduced gut transit time, yet co-housed mice did not, suggesting this outcome is not only malleable, but may be useful as a clinical readout for FD therapies.

The increased level of TMAO observed in FD patient serum is noteworthy given the positive predictive value of this metabolite for cardiovascular and kidney diseases[31,32]. Both diseases are leading causes of death among FD patients and together are the most

common non-acute cause of death[3] (acute forms being sudden death during sleep, aspiration, and complications due to aspiration, like pneumonia and sepsis). TMAO can directly alter neuronal physiology, such as increasing brain aging in mice by increasing the senescence of primary neurons[44]. As recently reviewed[45], TMAO is currently the most important microbial metabolite associated with Alzheimer's Disease and increased levels in cerebrospinal fluid was recently shown to be associated with dementia and pathology[46]. Future studies are needed to directly address whether TMAO accelerates neurodegeneration of the peripheral nervous system and whether decreasing TMAO levels is therapeutic. For example, dietary interventions like the choline-restricted diet used to treat the genetic disorder, trimethylaminuria (fish odor syndrome)[47], may help modify TMAO-mediated pathologies.

Elp1 is one of six subunits of the elongator holoenzyme, and in addition to the *ELP1* mutation giving rise to FD, polymorphisms in four other subunit loci have been associated with neurologic diseases, including intellectual disability (*ELP2, ELP4, ELP6*), ALS (*ELP3*), and ASD and Rolandic epilepsy (*ELP4*)[48–52]. Whether these non-*ELP1* mutations in the elongator complex influence the gut-metabolism axis has not been experimentally addressed to our knowledge, but therapeutic modification of the microbiome via fecal microbiome transplantation (FMT) has been evaluated in ASD and found effective for ameliorating both short- and long-term outcomes and both GI and neurologic symptoms[23,24,53]. Our results indicate that the progressive nature of *Elp1*-driven microbiome-metabolome dysbiosis is modifiable, and when controlled, can ameliorate deficits in gut function and pathology. We believe this finding provides rationale for clinical interventions like FMT that has been shown to prevent or lessen debilitating symptoms of other neurologic diseases, including ASD. Our results also suggest that gut function, microbiome diversity, and metabolic function may be clinically useful as empirical indicators of FD disease severity. Finally, given the fact that variants in other Elongator complex subunits have been associated with human cognitive function, ALS, ASD and/or epilepsy[48–52], our results may have broader therapeutic impacts for more common neurologic disorders.

## Methods

### Human subjects

Samples were obtained from enrolled participants under Institutional Review Board approved protocols at both Montana State University and the New York University School of Medicine (NYU Langone Health). FD patients and their relatives were recruited in a case-control design through the Dysautonomia Center at New York University (NYU) Langone Health. Patients were approached at annual clinic visits to NYU Langone Medical Center. If interested, cohabitating relatives were also approached about the study. All subjects (patients and relatives) were recruited with informed consent following IRB-approved protocols. The potential for self-selection bias was deemed minimal due to the genetic predisposition and highly penetrant nature of the disease. The Dysautonomia Center at NYU Langone serves the majority of FD patients around the world and given the prevalence of the disease in the Ashkenazi Jewish community, we assumed minimal potential of bias due to sample size. Participants received no compensation for participation in this study.

A single relative was enrolled for 27 FD patients; two relatives were enrolled for 13 patients; three relatives were enrolled for 1 patient; and no relatives were enrolled for 7 FD patients. All human samples were collected through the Dysautonomia Center at NYU Langone Health. Patients and relatives were not required to fast prior to any sample collection due to the fragile condition of some patients. Peripheral venous blood samples were collected in test tubes containing clot activator, allowed to coagulate for 30 min, and centrifuged at 1,000–2000× *g* for 10 min for serum separation. Serum was then aliquoted into 15 mL vials and frozen at −20 °C until it was shipped

overnight on dry ice to Montana State University where it was stored at −80 °C. Stool samples were self-collected using collection kits (commode/hat) containing gloves and a sterile tongue depressor and placed into a sterile 50 mL conical tube. These samples were requested to be collected as close as possible to blood samples. Tubes were sealed in zip-lock bags and frozen at −20 °C within 24 h. For most subjects, stool collection kits were sent to enrollees just prior to scheduled annual clinic visits so they could bring samples with them to their visit. Some subjects were recruited (potentially during these visits) and a stool collection kit was sent home. Samples collected at home were immediately frozen for at least 12 h and shipped overnight on ice and inside a small Styrofoam cooler box to the clinical team at NYU Langone. Samples were kept frozen at NYU (−20 °C), batched, and shipped overnight on dry ice to Montana State University and stored at −80 °C until processing. Patient metadata were pulled from medical charts or clinical laboratory values and evaluated as categorical predictor variables (Supplementary Table 2 in Supplementary Data 1). Pairing of patient and relative samples was done as much as possible with the only exclusion criterion being if samples were collected >90 days apart. All possible pairs were considered in analyses using permutation analyses (see Statistics below).

### Mice

All mouse experiments were conducted in the AALAC-accredited Animal Resource Center at Montana State University under local IACUC-approved protocols (MSU protocol no. 2021-35-81). Specific Pathogen Free (SPF) C57BL/6 mice carrying the *Tuba1a-Cre⁺; Elp1^loxp/loxp* mutation were used as the disease model (FD mice) and *Tuba1a-Cre⁻; /Elp1^+/loxp* littermates were used as controls. The creation and description of these genotypes have been described[34]. All experiments included both sexes and mouse age ranged from 21 days (0 days post-weaning; DPW) to 485 days (465 DPW) of age. Mice were housed under a 14-h light/10-h dark cycle with temperatures of 18–23 °C and 40–60% humidity, with food (irradiated PicoLab® Rodent Diet 20, 5053 Lab-Diet®, Land O'Lakes, Inc., Arden Hills, MN, USA) and water given ad libitum. To avoid cage effects, mice from at least two different cages per experimental group were analyzed. To test the effect of housing, groups of FD mice were either separated from littermate control siblings at weaning or left together.

For stool collection, mice were individually scruffed and freshly voided stool was collected into a sterile microcentrifuge tube, flash frozen in an ethanol dry ice bath, and frozen at −80 °C. To quantify disease in mice, a scoring system was devised based on eight factors (hind limb clasping, evidence of grooming, presence of cataracts, presence of kyphosis, motor function/movement, presence of tremors, observed jumping activity, and body condition) and scored via ordinal bins (Supplementary Table 3). Scores ranged from 0 (no disease) to 12 (severe disease). To quantify gut transit time, mice received an oral gavage of 150 μL of a solution comprised of 6% carmine red dye and 0.5% methylcellulose in water. A white paper towel was place on the floor of cages and mice were individually monitored every ten minutes for evidence of red stool. Gut transit time was calculated as the time between gavage and the presence of red stool.

### Stool microbiome sequencing

DNA was extracted from stool (human and mouse) using a DNeasy Powersoil kit (Qiagen, Hilden, Germany) following the manufacturer's instructions. DNA was submitted for 16 S rRNA gene sequencing at the University of Michigan Center for Microbial Systems (dual-index, paired-end sequencing[54]; 250 bp reads of variable region 4). Raw reads (forward and reverse) were processed using mothur[55] v.1.46.1 and assembled into contigs following published protocols[29,54]. Contigs with ambiguous bases or homopolymers of ≥8 bp were discarded. Identical sequences were combined and aligned to the V4 region of the 16 S rRNA alignment of the SILVA database (version

128, https://www.arb-silva.de/documentation/release-128). Chimeric sequences were identified with VSEARCH[56,57] and discarded. Contigs were classified within mothur using the Ribosomal Database Project's Bayesian classifier (training set 16, https://sourceforge.net/projects/rdp-classifier/files/RDP_Classifier_TrainingData). Contigs that classified as mitochondria, chloroplasts, or Eukaryota or were unclassified at the domain level were discarded. The remaining contigs were clustered into operational taxonomic units (OTUs) at ≥97% sequence identity using VSEARCH[57]. OTUs represented by fewer than 100 reads in the data set were removed. Human and mouse samples were rarefied to 9,848 and 5,308 reads, respectively, corresponding to the sample in each dataset with the least number of reads.

Partitioning around medoids (PAM) clustering was conducted using cluster analysis (cluster, version 2.1.3[58]) in R (version 4.1.0). Alpha diversity was estimated using the inverse Simpson index and ordinations (nonmetric multidimensional scaling; NMDS) were used to visualize between beta-diversity (Bray-Curtis dissimilarity). Diversity estimates were generated in R using vegan (version 2.6-2)[59], labdsv (version 2.0-1)[60], and custom scripts. Significance between groups in ordinations was evaluated using permutational multivariate analysis of variance (PERMANOVA) with the Adonis function in vegan and 9999 permutations.

### Preparation of serum and stool samples for metabolomics

Serum samples stored at −80 °C were thawed on ice and 400 μL serum was added to 1600 μL acetone (4:1 v/v ratio). Samples were mixed by inversion and incubated at room temperature (25 °C) for 20 min, followed by a second incubation at −20 °C for 1 h and centrifugation at 10,000× l for 10 min. Supernatant was transferred to a microcentrifuge tube and dried using a speed vacuum concentrator overnight, without heat. Dried metabolite mixtures were then reconstituted in 600 μL of serum NMR buffer consisting of 0.25 mM sodium 2, 2-Dimethyl-2-silapentane-5-sulfonic (DSS) in 90% $H_2O$/10% $D_2O$, 0.4 mM imidazole, 25 mM $NaH_2PO_4$/$Na_2HPO_4$, pH 7, and transferred to a 5 mm Bruker NMR tube.

Stool metabolite extraction from human samples was performed using methods adapted from previous publications[61,62] and is described in detail here. Samples were thawed at room temperature and homogenized by spatula stir mixing in their original 50 mL collection tubes. Aliquots of 500 mg were placed into 2 mL twist cap centrifuge tubes, flash-frozen and stored again at −80 °C. Further processing was done in batches. Samples were thawed and stool NMR buffer solution (0.1 M $Na_2HPO_4$/$KH_2PO_4$, pH 7.37, 0.25 mM sodium DSS, 0.4 mM Imidazole, 0.01% sodium azide, in 90% $H_2O$/10% $D_2O$, pH 7.0) was added at a 1:2 w/v ratio (wet stool-weight to buffer-volume). Homogenization and lysing were carried out using a bead-beating tissue lyser (Millipore FastPrep-24™ 5 G bead beating tissue lyser set at 6.0 m/s for 40 s), followed by centrifugation at 10,000× g for 10 min. Resulting supernatants were transferred to a new 2 mL centrifuge tube and the pellet subjected to a subsequent round of extraction. Supernatant from two rounds of bead beating per sample were combined and centrifuged at 21,000× g for 10 min, followed by filtration (4 μm syringe filter), followed by a second centrifugation step (21,000 × g for 10 min). The resulting solution was filtered using a 3 kDa filter to remove additional large particulates, and the final pH was adjusted to 7.0. 600 μL of filtered supernatant was transferred to a 5 mm Bruker NMR tube for NMR analysis. A separate aliquot of 500 mg of each human stool sample was lyophilized to determine water-weight percent, and dry weight of extracted samples. Mouse stool samples were prepared similarly to human stool samples with the following exceptions: NMR Buffer was added to achieve a 1:10 w/v (40–180 mg wet stool-weight to buffer-volume) ratio; samples were filtered with only a 3 kDa filter; 600 μL of filtrate was dried overnight under vacuum without heat and resuspended in 600 μL of 90% $H_2O$ to 10% $D_2O$.

### NMR-based metabolomics

All NMR spectra were collected at 300 K using a Bruker 600 MHz ($^1$H Larmor frequency) AVANCE III solution NMR spectrometer equipped with a 5 mm triple resonance ($^1$H, $^{13}$C, $^{15}$N) liquid helium-cooled cryoProbe, automatic sample loading system (SampleJet), and TopSpin software (Bruker version 3.2). Spectral acquisition of 1D $^1$H NMR experiments was performed using the Bruker gradient-based water suppression "zgesgp" pulse sequence[1,2], and 1D 1H NMR spectra were recorded using 256 scans, a $^1$H spectral window of ~12 ppm, 64 K data points, and a dwell time of 69 microseconds between points, resulting in an FID (free induction decay) data acquisition time period of ~4.5 s. Recovery delay time (i.e., D1 parameter) between acquisitions was set to 2 s, resulting in a total relaxation recovery delay of ~6.5 s between scans. DSS chemical shift referencing and phase correction of 1D $^1$H NMR spectra were conducted using TopSpin (Bruker version 3.2)[63]. Additional manual processing of 1D $^1$H NMR spectra and metabolite profiling were conducted using the Chenomx NMR Suite software (version 8.4; Chenomx Inc., Edmonton, Alberta, Canada). Baseline correction of NMR spectra following an import of preprocessed '1r' NMR spectral files into the Chenomx software was performed using the automatic cubic spline function (Chenomx Spline) in Chenomx, and subsequent manual breakpoint adjustment to obtain a flat, well-defined baseline, following guidelines from Chenomx application notes and reported methods[63–65]. $^1$H chemical shifts were referenced to the most upfield signal of DSS, set to 0.0 ppm, and the $^1$H NMR resonance of imidazole was used to correct for small chemical shift variations arising from slight changes in sample pH. Metabolite identification and quantitation were performed by fitting 1D $^1$H spectral splitting patterns, chemical shifts, and spectral intensities to reference spectra of small molecules accessible through the Chenomx small molecule spectral database for 14.1 T (600 MHz $^1$H Larmor frequency) magnetic field strength NMR instruments and the human metabolome database (HMDB). In addition, small spectral adjustments were made using a manual peak-based fit style in Chenomx to achieve optimal fits for compound peak cluster location and spectral intensity matching[66]. Relative intensities were calibrated to DSS concentration (0.25 mM), which was used as an internal reference to quantify metabolite levels and to determine metabolite concentrations. Confirmation of metabolite ID was achieved by spiking of pure metabolite standards into samples, as needed. Metabolite profiles were exported from the Chenomx software and converted from μM to nanomoles/mL for serum, nanomoles/gram for human stool, and nanomoles/mg for mouse stool by normalizing to serum volume, fecal dry mass for human stool, and fecal wet mass for mouse stool, respectively, and taking into account the NMR sample volume (600 μL). Metabolite data were log-transformed and auto-scaled (mean centered and divided by standard deviation) with mouse stool data being additionally normalized by sum, using MetaboAnalyst 4.0[67].

Metabolomic diversity (i.e., the presence/absence and abundance of metabolites) was visualized using ordination (partial least square-discriminant analysis, PLSDA; principal coordinates analysis) in R version 4.1.0 using vegan[59] (version 2.6-2), vegan3d[68] (version 1.1-2), labdsv[60] (version 2.0-1), mixOmics[69], and custom scripts. Axes in ordinations were labeled in ascending order corresponding to the amount of described variation (greatest to least). The validity of the PLSDA models were assessed using MetaboAnalyst version 3.0 and custom R scripts with a leave-one-out validation test for predictive variability ($Q^2$) metrics and predicted residual sum of squares (PRESS) error ($R^2$), prediction accuracy and separation distance permutation tests (n = 1000), classification error rates (CER), and area under the receiver operating characteristic curve (AUROC) analyses[70].

### TMA and TMAO analysis LCMS

Human serum samples were thawed at room temperature and 20 μL was added to 80 μL of −20 °C acetone in a clean vial and incubated

overnight at −80 °C to precipitate protein[71]. Following centrifugation, the metabolite-containing supernatant was removed and concentrated to dryness using an Eppendorf Concentrator Plus (Eppendorf, Hamburg, Germany) at 30 °C for ~30 min. 50 μL of MeOH:$H_2O$ (50:50) was added to the dried layer and vortexed to reconstitute the metabolite fraction. The sample was split into two 25 μL aliquots for each targeted LCMS analysis.

Stool metabolites were extracted for LCMS analysis using standard methods[72]. Approximately 500 mg of stool was thawed at room temperature and kept on ice for the remainder of the extraction. Three volumes of HPLC-grade water were added to stool samples, vortexed for 30 s, and homogenized using an Ultrasonic Homogenizer 3000 (Biologics, Manassas, VA) set at a 40% duty cycle for five minutes. An equal volume of MeOH, ~1 mL, was added to the slurry, vortexed for 30 s, and centrifuged (20,000 × g for 30 min). The metabolite-containing supernatant was removed and the cell debris pellet was washed. Washes consisted of the addition of MeOH:$H_2O$ to cover the debris pellet and agitation via vortexing. Following centrifugation (20,000 g for 30 min), the supernatant was removed and added to the previously collected supernatant. Acetone precipitation was conducted overnight (5 volumes of −20 °C acetone stored at −80 °C) and the supernatant was filtered using a 3Kda spin filter. Eluent was concentrated to dryness (Eppendorf Concentrator Plus at 30 °C for 2–3 h) and dried samples were reconstituted in 100 μL of MeOH:$H_2O$ (50:50) when ready for LCMS analysis. The sample was divided into two 50 μL aliquots, one for each targeted LCMS analysis.

One of the fecal metabolite extracts from each original sample was derivatized with dansyl chloride for TMA analysis[73]. To begin, 2 μL of 166 mM NaOH was added to each extract and the pH was verified to be between 9 and 9.5 using a pH probe. To this solution, 46 μL of 20 mg/mL dansyl chloride was added and incubated at room temperature for 30 min. After 30 min, 2 μL of 10% formic acid was added to quench the reaction and the pH was verified to be ~4 using a pH probe. Derivatized samples were placed in a clean vial for LCMS analysis.

LCMS analysis for derivatized and underivatized serum and fecal samples was completed on a Waters Synapt-XS Q-IMS-TOF (quadrupole-ion mobility spectrometry-time of flight) mass spectrometer coupled to a Waters I-Class UHPLC (ultra-high-performance liquid chromatography) (Waters, Milford, MA). Separation was achieved with a 130 Å, 1.7 um, 2.1 mm × 100 mm Acquity BEH-HILIC column in a column compartment kept at 30 °C. 5 μL of sample was injected and ionized using electrospray ionization in positive mode. Total method runtime was 6 min with mobile phase A consisting of 15 mmol/L ammonium formate while mobile phase B was acetonitrile. The gradient used was 10% A from 0–2 min, 10–40% A from 2–3.5 min, 40% A from 3.5–4.5 min and 10% A from 4.5–6 min[74]. Authentic standards were used to verify the m/z value and retention time as well as create standard curves for all analytes. Data analysis and normalization was completed in Progenesis QI version 3.0 and statistical analysis was performed in R version 4.1.0.

### Statistics
No statistical methods were used to predetermine sample sizes. Data were visually assessed for normality with quantile-quantile plots and, when possible, transformed for normality. The exact number ($n$) of replicates used in each experiment are reported in the respective figure legends. Statistical analyses were conducted in R version 4.1.0. PERMANOVA was used to test for significant difference observed between patients and relatives in ordinations (NMDS and principal coordinates analysis) or PAM clustered defined enterotypes (R package cluster, version 2.1.3). Random forest analysis (R packages randomForest version 4.7-1[75] and rfutilities version 2.1-5[76]) was used to identify both OTUs and individual metabolites responsible for significant differences. Visualization tools in R, including gplots (version 3.1.3)[77], RColorBrewer (version 1.1-3)[78], and dplyr (version1.1.10)[79], were

used to generate figures. Paired relative samples were not available for all FD patients and for other FD patients, multiple relatives provided samples within the required 90-day window (i.e., samples outside of this window were excluded). To leverage as many samples (and statistical power) as possible, pairs were randomly selected during permutation analyses performed in R. The number of pairs selected during each permutation was equal to the total number of patients for which a relative sample was available (FD patient samples without a paired relative sample were excluded from paired permutation analyses). FDR-corrected $p$ values were generated for all tests with multiple comparisons as described in the respective figures legends. Linear mixed-effects models were generated and evaluated using R package lme4, version 1.1–29[80]. Data distribution was evaluated and displayed for some factors using violin plots (R package vioplot, version 0.3.7[81]).

### Reporting summary
Further information on research design is available in the Nature Portfolio Reporting Summary linked to this article.

### Data availability
All sequencing reads generated in this study have been deposited in the National Center for Biotechnology Information (NCBI) BioProject database under accession number PRJNA785599. All metabolomics data generated in this study have been deposited in the MetaboLights repository under accession number MTBLS5138 and are also available at GitHub (https://github.com/nvpinkham/Dysautonomia). The data and analyses generated in this study are available as a single compressed Source Data file on GitHub (https://github.com/nvpinkham/Dysautonomia). The Silva 16 S rRNA database used for alignment is available at https://www.arb-silva.de/documentation/release-128 and the RDP training set used is available at https://sourceforge.net/projects/rdp-classifier/files/RDP_Classifier_TrainingData. Microbiome (16 S rRNA sequencing) datasets used for comparisons are available at the NCBI BioProject database accession numbers PRJNA505353 (Martinson et al.) and PRJNA290926 (Baxter et al.). Source data are provided with this paper.

### Code availability
Custom code was used to generate figures and statistical comparisons has been deposited in the Zenodo repository (https://doi.org/10.5281/zenodo.7384390) and is available at GitHub (https://github.com/nvpinkham/Dysautonomia)[82].

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

## Acknowledgements

Research reported in this publication was supported by the National Institute of Diabetes and Digestive and Kidney Diseases of the National Institutes of Health under award number R01DK117473 to V.C., F.L., and S.T.W.; a supplement for graduate education R01DK117473-02S1 to V.C., S.T.W, and S.M.C.; and support from the National Institute of General Medical Sciences of the National Institutes of Health under award number P20GM103474 to S.M.C. The content is solely the responsibility of the authors and does not necessarily represent the official views of the National Institutes of Health. Additional graduate training support to S.M.C. was provided by the Alfred P. Sloan Foundation and the Montana INBRE Native American Graduate Fellowship by the National Institute of General Medical Sciences of the National Institutes of Health under Award Number P20GM103474. The content is solely the responsibility of the authors and does not necessarily represent the official views of the National Institutes of Health. Additional graduate stipend and training support for A.M.C. from the Molecular Biosciences Program at Montana State University (MSU). Funding for the Proteomics, Metabolomics and Mass Spectrometry Facility used in this publication was made possible in part by the MJ Murdock Charitable Trust, the National Institute of General Medical Sciences of the National Institutes of Health under Award Numbers P20GM103474 and S10OD28650, and the MSU Office of Research, Economic Development and Graduate Education to B.B., S.M.C., and V.C. Financial support for the NMR instruments and operations of MSU's NMR Core Facility has been provided in part by the NIH Shared Instrumentation Grant (SIG) program (1S10RR13878 and 1S10RR026659), the National Science Foundation (NSF-MRI:DBI-1532078; NSF-MRI:CHE–2018388), the Murdock Charitable Trust Foundation (2015066:MNL), and MSU's Office of the Vice President for Research and Economic Development to V.C. We thank Galen O'Shea-Stone, Christina Lucas, Lynn George, Martha Chaverra, and Vickie Riojas for technical support and all the families that provided samples for this study.

## Author contributions

A.M.C., S.M.C., and N.V.P. contributed equally to this work, including acquisition, analysis and interpretation of data and drafting of this manuscript. N.V.P. generated new software used to analyze results. A.W. contributed to data acquisition (sample preparation and metabolite profiling using NMR). S.C.B. and M.C.-V. contributed to sample and data acquisition, including obtaining samples from FD patients and their relatives. M.M., D.J.K., and H.M.G.-W. were involved in experimental design, data acquisition, and drafting of this manuscript. B.T., J.P., and B.B. contributed to data acquisition and analysis and drafting of this

manuscript. H.K. and L.N.-K. contributed to all clinical aspects of the study, including data acquisition and interpretation of results, and drafting of this manuscript. F.L., V.C., and S.T.W. contributed to all aspects of the study and led the team's efforts in overall conception of the project, data analysis and interpretation, and drafting of this manuscript.

## Competing interests

The authors declare no competing interests.
