## [Peer Review File · Nature Communications]

REVIEWER COMMENTS

Reviewer #1 -Microbiome / Neurodegenerative disease / Metabolomics- (Remarks to the Author):

In this work, the authors analyze the fecal microbiome and metabolome along with the serum metabolome from a nice sampling of individuals with FD and appropriately matched controls. This was also performed in an FD mouse model, where homogenizing the gut microbiome via cohousing the animals helped to ameliorate the observed effects. The primary finding in the metabolomics is that high levels of choline are measured in FD patients. The manuscript is well-written and the data are presented clearly.

Specific comments:

Text:

In the context of a disease with an almost absolute genetic penetrance, one may question why a metabolomic, microbial, environmental approach would be explored. It seems more obvious to explore gene therapy or genetic screening, so a brief addition of text might make the manuscript stronger. For instance text describing the state of genetic options, the lack of options for therapy for those currently living with the condition, and the potential applications of these findings to a broader context of other neurological diseases with this work almost as a proof of principle, may increase the impact and relevance of the work to a broader audience. This was done very nicely in the discussion. Perhaps add a brief sentence to the introduction to offer this context from the start.

The controls in the study are likely to be carriers of the risk allele- is it possible that they have intermediate levels for some metabolites between “disease” and “healthy” levels, making you miss something? Have homozygous WT control individuals ever been compared to FD patients? Are these subclasses that could be compared in the current study?

The use of the term “less homogeneous” to describe a higher beta-dispersion could be rephrased if possible. Currently, for instance on line 70, the terms “less diverse” and “more homogeneous” sound contradictory within the same sentence. If I am understanding correctly, the former refers to diversity within the samples and the latter refers to the spread of data within the FD patient group. However, this is not immediately clear from the current wording.

Please mention in the main text how many metabolite species were identified in fecal and serum samples in total.

Experimental:

Do FD mice fed a high-choline diet (as well as controls) fare even worse than the standard genetic line?

The use of a g-bag in a subset of FD patients is an extremely large caveat that should greatly influence metabolism and fecal microbiome. Can you also do the analysis leaving these patients out (or stratifying into subgroups) and see if the same metabolites seem to be of interest?

Figures:

Please use a consistent color scheme throughout the figures. For instance, where logical, all FD vs control data can be the same colors throughout the entire set of figures.

In the data presented in figure 4a and b, why are earlier timepoints not included as done in the previous figure? For instance- at day 14 FD mouse microbiome has already diverged but prior data is not shown.

Figure 4d, when these results are analyzed separately for each DPW age, is it still significant? Is a certain time point the strongest? It seems unusual to present all time points collapsed into one metric.

Figure 4c, the time range is really broad- are the groups balanced within this window? Almost 100 days span is a long time in a mouse. Why is the data acquired and presented this way? Were there multiple testing days and scores were averaged for each mouse in a balanced way?

Discussion:

Are parallel observations occurring in the ELP1 mutant in yeast? You mention differences, are any of them relevant?

In this sentence, clarify if you are speculating that this work indicates that this will be possible in the context of other diseases or for FD specifically. To my understanding, neither is needed with

diagnosis of FD “At a minimum, our results suggest that gut function and microbiome-metabolome diversity can be used clinically or in therapeutic trials as biomarkers of disease.”

Manuscript ends rather abruptly while discussion of a tangent. Consider adding a sentence or two in general conclusion.

Reviewer #2 -Microbiota-brain axis / neurological diseases- (Remarks to the Author):

Familial dysautonomia (FD) is a genetic disorder that affects nerve cells in the autonomic nervous system, including the development and survival of sensory, sympathetic, and parasympathetic neurons. The consequences are visible in symptoms such as gastrointestinal dysfunction, altered pain sensitivity, altered temperature perception, blood pressure instability as well as frequent developmental delays. In this study, the authors investigate a potential role of the gut microbiome in FD, evaluating both human subjects and in the neuron-specific ELP1-deficient mouse model of FD. The main finding reported is that FD is associated with significant differences in gut microbiome composition and function, assessed using 16S rRNA sequencing and ¹H NMR-based metabolomics. The authors were also able to improve deficits in gut transit and reduce disease severity by cohousing mutant and littermate control mice.

This is a potentially important study and I welcome the focus on a rare disorder. It benefits from a translational approach, drawing together information from human and a relevant animal model. It is also notable that the authors look at both microbiome composition and function. I have the following comments and queries:

(1) The abstract states that key observations in humans were largely recapitulated in the ELP1-deficient mouse model. Could the authors elaborate some more on this point since the gut microbiome and metabolome analysis of the samples taken from the mouse model do not really go into that much detail on this point. It seems that this is largely based on the microbial diversity analysis from the 16S sequencing data, and while metagenomic profiling might be more informative in this regard, are there similar compositional alterations at the genera/species level? There are similar questions around the co-housing experiment in terms of engraftment of the control microbiota, and associations between the microbiota/metabolites and improvements in pathology.

For example, did any of the metabolites implicated from the human study vary in production between cohoused and separately housed animals?

(2) The implications of the results are often overstated in the discussion. Although I agree that the possibility of targeting the microbiota to improve deficits in gut function and pathology is appealing, there is some way to go before the results presented here can be 'used clinically or in therapeutic trials as biomarkers of disease'. It is also premature at this early stage to advocate for FMT as a clinical intervention, especially on the back of this study and small open label studies in ASD. A revised discussion should avoid overselling the results in this way.

(3) Some key experiments are missing, such as FMT from human to rodent, to understand more accurately the implications of the FD-associated microbiota. It is not really that surprising, given the symptom profile and other characteristics of FD patients, that there are microbiome alterations at the compositional and functional level. I think then that the key question the authors ask is whether the FD-associated microbiota, once established, perpetuates further neurodegeneration and symptom expression. A more detailed and nuanced analysis of the mouse microbiome and metabolome is required to really conclude that the microbiome and/or metabolite changes in the cohousing experiments are linked to improvements in pathology. The suggested involvement of specific metabolites is also a testable hypothesis that would greatly increase the value of this work.

Reviewer #3 -Familial Dysautonomia / Neurodegenerative disorders- (Remarks to the Author):

Summary:

Cheney et al. present an intriguing set of data showing that in the neurodevelopmental and neurodegenerative disorder FD the gut microbiome and metabolome is different from healthy relatives. The data is supported by similar findings in the FD mouse model. FD mice pathologies are reduced by co-housing, a natural form of fecal matter transplant in mice, which lends important hope that FMT should be further pursued for treatment of FD patients. The overall finding that neurodegenerative disease negatively impact the gut microbiome axis is important and supported by previous findings in other neurodegenerative disorders. However, specifically in the PNS this is new and a crucial piece of knowledge for the understanding of various disorders as well as for the moving forward with currently lacking treatments for FD and other patients.

Strength:

- The co-founding factors that might/are affecting microbiome and metabolite diversity in FD patients, including age, feeding tube are well discussed. FD is not reported to have a sex differences however, it might still be interesting to include a sex analysis here with this existing data.
- Fig 4c is an important finding with respect to patient care and lends hope that fecal transplant approaches might be successful on more than one pathology level.
- Fig 4d is another important finding with respect to patient care. With respect to this finding, have enteric neurons specifically been investigated for disrupted function in FD before or is it mainly a neuronal numbers question?

Major weaknesses:

- The overall finding here that FD patient's microbiome/metabolome is different from their relatives, have similar findings been reported in other neurodegenerative disorders? Has this ever been investigated before?
- Fig 3b. what is light and dark grey here?
- Fig 4 a, b representation of the data is difficult to understand, essentially I don't understand the graph and that leads to it being hard to follow how the authors came to their conclusion. Maybe one could use 4 colors, ie. light green for FD co-housed and dark green for ctrl cohoused, and light brown for FD separate housed and dark brown for ctrl separate housed? Also, why are cohoused animals at day 179 and 279 the same again? Is this suggesting that FMT may be given for 79 days and even if it is stopped after the positive effects continue? If so, somehow the data needs to be presented in a simpler way for the reader to be able to follow.
- Line 167, 'a dysfunctional gut-metabolism axis that promotes pathology reminiscent of more common neurodegenerative diseases and other neurologic/neuropsychiatric conditions.' It would be good to expand on this statement with more specifics and literature examples. The overall finding of this paper, that the gut-microbiome axis is disturbed in neurodegenerative disease (especially in the PNS) is very important and new. It would be great to put this finding in relation to more common disorders to show the impact of the author's findings to neurodegenerative disorders more broadly.

Minor Weaknesses:

- Fig 1a, the connecting lines are useful, but most are hard/impossible to see. For example, do patients with small diversity mostly also have relatives with small diversity? Maybe one could highlight a few such connections to be able to see that better.
- Extended data 5 figure is too small to read.
- Line 181-184 this statements should be discussed much earlier in the manuscript.

- Line 201: remove 'At a minimum'
- Line 551-552 should be mentioned in the manuscript proper

Other considerations:

- Data analysis, interpretation and conclusions seem solid. Several graphs are not easy to understand for someone who is not an expert in microbiome and metabolomics and statistics thereof, if there are ways to simplify or explain a bit more that would be beneficial.
- The methodology is sound and the methods seems detailed enough to good reproducibility.

Response to reviewers' comments

We thank the reviewers for their positive and constructive comments and for their appreciation of our study. We hope they will find our responses to their concerns satisfactory in this revised manuscript.

We wish to draw reviewers' attention to the removal of the human serum metabolite, 4-pyridoxate, from our results. Signal peaks for all metabolites were re-checked against compound standards, either using spectra from validated metabolite databases (HMDB, Chemomx library) or directly via analysis of chemical standards. In doing so, peaks used to identify 4-pyridoxate did not overlap with the chemical standard, which we did not have before the original manuscript was submitted. Resonances for this compound overlap with the signals of other compounds that we had already validated, leading us to conclude that this metabolite was originally mis-identified. Fortunately, removal of this metabolite did not affect the primary results or conclusions of our study. We only wish to bring this to your attention given this revision to our original manuscript (Fig. 2b and Supplementary Fig. 5a).

Reviewer #1:

Comment: In this work, the authors analyze the fecal microbiome and metabolome along with the serum metabolome from a nice sampling of individuals with FD and appropriately matched controls. This was also performed in an FD mouse model, where homogenizing the gut microbiome via cohousing the animals helped to ameliorate the observed effects. The primary finding in the metabolomics is that high levels of choline are measured in FD patients. The manuscript is well-written and the data are presented clearly.

Specific comments:

Text:

In the context of a disease with an almost absolute genetic penetrance, one may question why a metabolomic, microbial, environmental approach would be explored. It seems more obvious to explore gene therapy or genetic screening, so a brief addition of text might make the manuscript stronger. For instance, text describing the state of genetic options, the lack of options for therapy for those currently living with the condition, and the potential applications of these findings to a broader context of other neurological diseases with this work almost as a proof of principle, may increase the impact and relevance of the work to a broader audience. This was done very nicely in the discussion. Perhaps add a brief sentence to the introduction to offer this context from the start.

Response: We appreciate this suggestion and in response, have added relevant text to the introduction. We also cited a recent review of GI symptoms in FD to provide additional rationale for our study's focus on the gut microbiome-metabolome.

Comment: The controls in the study are likely to be carriers of the risk allele- is it possible that they have intermediate levels for some metabolites between "disease" and "healthy" levels, making you miss something? Have homozygous WT control individuals ever been compared to FD patients? Are these subclasses that could be compared in the current study?

Response: This is an excellent suggestion, and in response we have included a relevant microbiome comparison in our revision (Results section and Supplementary Fig. 4). Briefly, we evaluated two previously published 16S rRNA sequencing datasets from healthy adults (Baxter

et al and Martinson et al). One of these datasets was generated in our laboratory and the other was not but used a nearly identical sequencing approach. The datasets allowed us to incorporate both large, cross-sectional and focused, longitudinal microbiome diversity of healthy adults. Results from these comparisons were interesting and support other studies that found it difficult to compare 16S rRNA sequencing studies due to significant study-to-study variability. Overall - and most important - the microbiomes of FD relatives were more similar to those of healthy subjects in these two other studies than they were to FD patient microbiomes, which adds confidence that patient microbiomes are truly different from “healthy” individuals. These new results also highlight the strength of our case-control (FD patient-FD relative) design, whereby matching was used as suggested by a recently published study in *Nature* (Vujkovic-Cvijin et al). Finally, these new data make sense given that being a heterozygote for the ELP1 FD mutation has been well documented not to affect health; such carriers are highly functional, perfectly healthy “typical” adults.

Regarding metabolomics data, while NMR is highly reproducible, comparisons between studies is not straightforward due to technical differences in sample preparation, instrumentation, and machine running parameters. Unfortunately, to test whether carriers of the ELP1 mutant allele display an intermediate metabolic phenotype (i.e., an NMR metabolome profile between FD patients and WT control subjects) will involve collection of a new set of samples, presumably from WT siblings of FD patients. Given that true WT would only comprise 25% of all patient siblings, this creates a rather difficult technical hurdle to overcome. While this comparison may indeed be interesting, we believe it is beyond the scope of the current project, especially since we found little evidence that the microbiomes of ELP1 mutant carriers differed from those of healthy individuals in two other studies beyond that expected from inter-study variability.

Comment: The use of the term “less homogeneous” to describe a higher beta-dispersion could be rephrased if possible. Currently, for instance on line 70, the terms “less diverse” and “more homogeneous” sound contradictory within the same sentence. If I am understanding correctly, the former refers to diversity within the samples and the latter refers to the spread of data within the FD patient group. However, this is not immediately clear from the current wording.

Response: We agree that these terms/concepts can be confusing. The revised text describes microbiome “variability between subjects” as a measure of dispersion, which hopefully adds clarity.

Comment: Please mention in the main text how many metabolite species were identified in fecal and serum samples in total.

Response: The total number of human serum and stool metabolites identified (n=55 and n=73, respectively) were reported in the original main text (Results: lines 85 and 87). We realize now that we did not list the number of stool metabolites that were identified and quantified in the cohoused mouse studies. This omission has been corrected, and we now mention mouse stool metabolites (n=68) in the results section.

Comment:

Experimental:

Do FD mice fed a high-choline diet (as well as controls) fare even worse than the standard genetic line?

Response: We thank the reviewer for the suggested experiment. Unfortunately, we have not yet exposed FD mice to alternative diets. This is an intriguing hypothesis requiring additional time/effort that we hope to get to in the near future.

Comment: The use of a g-bag in a subset of FD patients is an extremely large caveat that should greatly influence metabolism and fecal microbiome. Can you also do the analysis leaving these patients out (or stratifying into subgroups) and see if the same metabolites seem to be of interest?

Response: We agree this is an important factor to consider and have tried to clarify the impact of placement of a G-tube on microbiome-metabolome diversity in the revised text. There are a few issues with stratifying by G-tube (with vs. without G-tube). The most important is that most patients had a G-tube; only 6 (of 32) and 8 (of 33) patients did not have a G-tube in our serum and stool metabolite datasets, respectively. When we tested for differences in metabolite levels between these groups, no differences reached statistical significance after FDR correction, but this may simply be due to low sample size. In our revision, we've clarified use of G-tubes in FD patients and the fact that this factor covaries with age, making it difficult to independently evaluate. It is worth noting (here and in the revised text) that not all patients with a G-tube used it exclusively. In fact, the exclusive use of a G-tube was a very significant factor in microbiome diversity and its presence-absence was associated with metabolome (choline) diversity. This factor, however, is also confounded by age because as FD patients age, the likelihood that they use a G-tube exclusively increases due to the disease progression. Put simply, age and G-tube are confounded by the pathological progression of FD and require additional studies to differentiate their true interactions.

Comment:

Figures:

Please use a consistent color scheme throughout the figures. For instance, where logical, all FD vs control data can be the same colors throughout the entire set of figures.

Response: We see how the color scheme in figures may be confusing and have revised them where logical. Human groups are consistent throughout. Mouse groups are also consistent (e.g., separately housed mice are consistently colored between Figs. 3 and 4) and colors were changed to grayscale in panels where they were not needed.

Comment: In the data presented in figure 4a and b, why are earlier timepoints not included as done in the previous figure? For instance- at day 14 FD mouse microbiome has already diverged but prior data is not shown.

Response: Figures 4a and b describe microbiome and metabolome differences, respectively, at timepoints corresponding to when signs of disease become evident in FD mice (>3 months). Microbiome results shown in Figures 3a-c were shown at earlier timepoints to capture the progressive nature of divergence (i.e., non-significant at weaning but rapidly becoming different). These details have been clarified in the revision.

Comment: Figure 4d, when these results are analyzed separately for each DPW age, is it still significant? Is a certain time point the strongest? It seems unusual to present all time points collapsed into one metric.

Response: We appreciate this comment, and the short answer is: yes – the results are still significant. We've included DPW as a potential explanatory factor in a mixed effects model and a new figure in the supplement (Supplemental Fig. 7), showing both phenotype scores and gut transit times as continuous data instead of categorical as shown in Fig. 4c and 4d. Hopefully presenting results both ways helps clarify the observed trends and their significance.

Comment: Figure 4c, the time range is really broad- are the groups balanced within this window? Almost 100 days span is a long time in a mouse. Why is the data acquired and presented this way? Were there multiple testing days and scores were averaged for each mouse in a balanced way?

Response: This comment was addressed in the response to the previous comment and new figure in supplement (Supplemental Fig. 6).

Comment:

Discussion:

Are parallel observations occurring in the ELP1 mutant in yeast? You mention differences, are any of them relevant?

Response: While deletion of the *Elp3* subunit of the Elongator complex in yeast altered the metabolome (Kalsborn et al., 2016), none of the reported metabolic changes were the same as what we detected in FD patients or FD mice, which have reductions in *Elp1*. These differences were likely due to species differences in major metabolism pathways and/or the fact that it is still unresolved to what extent *Elp1* and *Elp3* might function independently, outside of their combined role in the Elongator complex. We've clarified this in the Discussion section.

Comment: In this sentence, clarify if you are speculating that this work indicates that this will be possible in the context of other diseases or for FD specifically. To my understanding, neither is needed with diagnosis of FD "At a minimum, our results suggest that gut function and microbiome-metabolome diversity can be used clinically or in therapeutic trials as biomarkers of disease."

Response: The microbiome/metabolome is not needed for FD diagnosis. The meaning here was that these empirical outcomes could be used as a correlate and biomarker of disease severity in general, and most interestingly, in response to therapeutic interventions. Of course, this idea remains hypothetical until explicitly tested. We removed the word "biomarker" to avoid confusion and revised this sentence to clarify our meaning.

Comment: Manuscript ends rather abruptly while discussion of a tangent. Consider adding a sentence or two in general conclusion.

Response: We agree and in response, have modified the Discussion ending.

Reviewer #2 (Remarks to the Author):

Comment: Familial dysautonomia (FD) is a genetic disorder that affects nerve cells in the autonomic nervous system, including the development and survival of sensory, sympathetic, and parasympathetic neurons. The consequences are visible in symptoms such as gastrointestinal dysfunction, altered pain sensitivity, altered temperature perception, blood pressure instability as well as frequent developmental delays. In this study, the authors investigate a potential role of the gut microbiome in FD, evaluating both human subjects and in the neuron-specific ELP1-deficient mouse model of FD. The main finding reported is that FD is associated with significant differences in gut microbiome composition and function, assessed using 16S rRNA sequencing and 1H NMR-based metabolomics. The authors were also able to improve deficits in gut transit and reduce disease severity by cohousing mutant and littermate control mice.

This is a potentially important study and I welcome the focus on a rare disorder. It benefits from a translational approach, drawing together information from human and a relevant animal model. It is also notable that the authors look at both microbiome composition and function. I have the following comments and queries:

(1) The abstract states that key observations in humans were largely recapitulated in the ELP1-deficient mouse model. Could the authors elaborate some more on this point since the gut microbiome and metabolome analysis of the samples taken from the mouse model do not really go into that much detail on this point.

Response: We are happy to elaborate and have clarified the similarity in observations between mice and human patients (Discussion). Mice recapitulated a progressive pathology with key features similar to that observed in FD patients. Although the microbiomes of mice and humans are very different (see Venn diagram; Supplementary Fig. 9), the overall effect of FD on diversity was similar (especially increased beta-diversity). Finally, an increased level of choline was observed in separately housed FD mice compared to FD mice cohoused with control mice on day 279 DPW (Supplementary Fig. 5a).

Comment: It seems that this is largely based on the microbial diversity analysis from the 16S sequencing data, and while metagenomic profiling might be more informative in this regard, are there similar compositional alterations at the genera/species level? There are similar questions around the co-housing experiment in terms of engraftment of the control microbiota, and associations between the microbiota/metabolites and improvements in pathology. For example, did any of the metabolites implicated from the human study vary in production between cohoused and separately housed animals?

Response: Microbiome analyses were conducted with operational taxonomic units (OTUs) defined as $\geq 97\%$ sequence identity at the V4 region of bacterial 16S rRNA encoding gene. This definition is a field standard and corresponds well with named bacterial species, thus our approach directly evaluated microbiome change at the bacterial species level. Shotgun metagenomic sequencing could have been used to accomplish the same goal (i.e., identify species) and with the added benefit of identifying gene content differences. Since we used NMR-based metabolomics to assess microbiome function, which is a more direct evaluation of phenotype than gene content, we did not consider metagenomic sequencing to be necessary. Metabolomic differences between separately and cohoused mice are shown at three time points throughout disease progression in Supplementary Fig. 5. Choline was the only metabolite whose levels were significantly different in both humans and mice (Day 279 DPW) and this point has been emphasized more strongly in our revision (Discussion). We also found that methylamine, another microbiome metabolite of choline produced coincidentally with trimethylamine, was elevated in separately housed FD mice compared to cohoused FD mice. This point has also been emphasized in the revised results and discussion sections.

Comment: The implications of the results are often overstated in the discussion. Although I agree that the possibility of targeting the microbiota to improve deficits in gut function and pathology is appealing, there is some way to go before the results presented here can be 'used clinically or in therapeutic trials as biomarkers of disease'. It is also premature at this early stage to advocate for FMT as a clinical intervention, especially on the back of this study and small open label studies in ASD. A revised discussion should avoid overselling the results in this way.

Response: The reviewers seem split on the impact of our results regarding FMT. We believe our findings are consistent with the use of FMT for improving GI symptoms and possibly slowing the progression of pathology in FD patients. It is important to note – there are only two clinically used microbiome therapies; antibiotics and FMT. Our evidence supports the latter. That said, we agree that this is the first study of the microbiome in FD and the first to provide pre-clinical (animal) evidence that FMT may be beneficial. Follow-up studies are certainly needed to corroborate and extend this work. We have modified our text accordingly in the revised discussion.

Comment: Some key experiments are missing, such as FMT from human to rodent, to understand more accurately the implications of the FD-associated microbiota. It is not really that surprising, given the symptom profile and other characteristics of FD patients, that there are microbiome alterations at the compositional and functional level. I think then that the key question the authors ask is whether the FD-associated microbiota, once established, perpetuates further neurodegeneration and symptom expression. A more detailed and nuanced analysis of the mouse microbiome and metabolome is required to really conclude that the microbiome and/or metabolite changes in the cohousing experiments are linked to improvements in pathology. The suggested involvement of specific metabolites is also a testable hypothesis that would greatly increase the value of this work.

Response: We agree that conducting human → mouse FMT would be an informative experiment and one we are planning on doing. Unfortunately, we were unable to conduct this experiment to date. We humbly suggest that the data we do present are sufficiently novel and that the mouse co-housing data (which is effectively a type of FMT) provide compelling evidence. Since we show here that the control microbiome can ameliorate disease pathology and function (enhanced intestinal motility) in a mouse model of FD, the rationale for testing this therapeutic avenue is now a key topic of discussion with FD clinicians on our team.

Reviewer #3 -Familial Dysautonomia / Neurodegenerative disorders- (Remarks to the Author):

Comment:

Summary:

Cheney et al. present an intriguing set of data showing that in the neurodevelopmental and neurodegenerative disorder FD the gut microbiome and metabolome is different from healthy relatives. The data is supported by similar findings in the FD mouse model. FD mice pathologies are reduced by co-housing, a natural form of fecal matter transplant in mice, which lends important hope that FMT should be further pursued for treatment of FD patients. The overall finding that neurodegenerative disease negatively impact the gut microbiome axis is important and supported by previous findings in other neurodegenerative disorders. However, specifically in the PNS this is new and a crucial piece of knowledge for the understanding of various disorders as well as for the moving forward with currently lacking treatments for FD and other patients.

Strength:

The co-founding factors that might/are affecting microbiome and metabolite diversity in FD patients, including age, feeding tube are well discussed. FD is not reported to have a sex differences however, it might still be interesting to include a sex analysis here with this existing data.

Response: We appreciate this suggestion and have added this factor to our analysis. Sex did not explain microbiome or metabolome diversity as noted in the revised Supplementary Table 2, which is consistent with the lack of reported sex differences in FD clinical phenotypes.

Comment: Fig 4c is an important finding with respect to patient care and lends hope that fecal transplant approaches might be successful on more than one pathology level. Fig 4d is another important finding with respect to patient care. With respect to this finding, have enteric neurons specifically been investigated for disrupted function in FD before or is it mainly a neuronal numbers question?

Response: We appreciate this question and have added text highlighting the currently available information on the function and abundance of neurons in FD (Discussion). That said, this information is restricted to the upper GI tract and appendix.

Comment:

Major weaknesses:

The overall finding here that FD patient's microbiome/metabolome is different from their relatives, have similar findings been reported in other neurodegenerative disorders? Has this ever been investigated before?

Response: Our study is the first to evaluate the microbiome and metabolome of FD patients. To our knowledge our study is also the first to link microbiome and metabolome alteration with a peripheral neurodegenerative disease. At least three microbiome and/or metabolomics studies of Alzheimer's Disease and Parkinson's Disease patients enrolled patients' relatives as control subjects (now cited in the revision) but we were unable to find any that leveraged a case-control (patient-relative) design in statistical analyses (i.e., relatives were included but not used in paired analyses). A case-control design was recently shown to be important for minimizing the effects of confounding host variables (environment, diet, genetics, other) in microbiome studies involving human subjects and we presume the same is true for metabolomics studies as well. These points and relevant citations have been added to the revised discussion section.

Comment: Fig 3b. what is light and dark grey here?

Response: We apologize for this omission and have added a key to Fig. 3b.

Comment: Fig 4 a, b representation of the data is difficult to understand, essentially I don't understand the graph and that leads to it being hard to follow how the authors came to their conclusion. Maybe one could use 4 colors, ie. light green for FD co-housed and dark green for ctrl cohoused, and light brown for FD separate housed and dark brown for ctrl separate housed? Also, why are cohoused animals at day 179 and 279 the same again? Is this suggesting that FMT may be given for 79 days and even if it is stopped after the positive effects continue? If so, somehow the data needs to be presented in a simpler way for the reader to be able to follow.

Response: We apologize for the incomplete description of the figure. Colors in Fig. 4a,b have been replaced by grayscale due to this and another reviewer's comment. Similarly, we've included alternate visualizations of results in the supplement (Supplementary Figs. 6 and 7) that hopefully clarify these comparisons.

Comment: Line 167, 'a dysfunctional gut-metabolism axis that promotes pathology reminiscent of more common neurodegenerative diseases and other neurologic/neuropsychiatric conditions.' It would be good to expand on this statement with more specifics and literature examples. The overall finding of this paper, that the gut-microbiome axis is disturbed in neurodegenerative disease (especially in the PNS) is very important and new. It would be great to put this finding in

relation to more common disorders to show the impact of the author's findings to neurodegenerative disorders more broadly.

Response: We appreciate this comment and agree that our findings are the first to demonstrate an altered gut microbiome/metabolome in a peripheral neurodegenerative disease. As stated in our response to the previous comment, we have now conducted a literature search and tried to compare our results with the current state of the field.

Comment:

Minor Weaknesses:

Fig 1a, the connecting lines are useful, but most are hard/impossible to see. For example, do patients with small diversity mostly also have relatives with small diversity? Maybe one could highlight a few such connections to be able to see that better.

Response: We apologize for the complexity of the connecting lines, which is a result of sample size and unfortunately is not easy to show clearly. The key point with the connecting lines is that they are long compared to lines that connect healthy relatives. The original Supplementary Fig. 1b also show connections between FD patient-relative pairs and respective Bray-Curtis (BC) distance. We realize this may not be much clearer, so we've added another panel to Fig. 1 (Fig. 1b) that summarizes BC distances explicitly and hopefully this further clarifies the intent of the connecting lines. The comment about alpha diversity is intriguing. When examined, however, it was significant. In the end, we mentioned the hypothesis and analysis in the text (H_a = the alpha diversity of an FD patient is correlated with alpha diversity of their healthy relative(s)).

Comment: Extended data 5 figure is too small to read.

Response: This figure (now Supplementary Fig. 8) has been enlarged considerably.

Comment: Line 181-184 this statements should be discussed much earlier in the manuscript.

Response: Since a review of GI complaints in FD patients was recently published, we have now added this citation with clarifying text to the introduction.

Comment: Line 201: remove 'At a minimum'

Response: This phrase has been removed.

Comment: Line 551-552 should be mentioned in the manuscript proper

Response: This point has been moved to the Introduction.

Comment:

Other considerations:

Data analysis, interpretation and conclusions seem solid. Several graphs are not easy to understand for someone who is not an expert in microbiome and metabolomics and statistics thereof, if there are ways to simplify or explain a bit more that would be beneficial. The methodology is sound and the methods seems detailed enough to good reproducibility.

Response: Where possible, we've tried to more clearly explain results shown in figures and avoid or clarify the meaning of jargon in the field. We appreciate the acknowledgement of our rigorous approaches.

REVIEWERS' COMMENTS

Reviewer #1 (Remarks to the Author):

I appreciate the response to reviewers' comments, and feel that the manuscript is stronger as a result. The authors added new, requested data analyses. I have no issues with the data that is presented, but I do regret that no further animal experiments were performed, which I feel are necessary to tie the metabolomics data to the story.

The work is novel in a rare condition, which is worthy of study both for those suffering from the condition as well as for proof of principle and potential application to other conditions. However, the work remains correlative aside from showing that shifts in the mouse model microbiome can alleviate symptoms. At this point in the microbiome field, I expect a publication in such a high impact journal to move beyond correlation of metabolites and into testing. Especially in this case, where a single key metabolite is the clear signal to test. I only requested one experiment to confirm that the choline observation plays a role in severity of symptoms or as a potential target for therapeutics.

Reviewer #2 (Remarks to the Author):

The concerns from the initial reviews have largely been reasonably addressed in the revisions

Reviewer #3 (Remarks to the Author):

All my concerns have been adequately addressed.

Response to Reviewers' comments on revised manuscript

We thank the reviewers for their comments on our revised manuscript. We are delighted that reviewers generally felt our response to primary review was reasonable and agree that the revised manuscript is much stronger than the primary submission.

Reviewer #1:

Comment: I appreciate the response to reviewers' comments, and feel that the manuscript is stronger as a result. The authors added new, requested data analyses. I have no issues with the data that is presented, but I do regret that no further animal experiments were performed, which I feel are necessary to tie the metabolomics data to the story.

The work is novel in a rare condition, which is worthy of study both for those suffering from the condition as well as for proof of principle and potential application to other conditions. However, the work remains correlative aside from showing that shifts in the mouse model microbiome can alleviate symptoms. At this point in the microbiome field, I expect a publication in such a high impact journal to move beyond correlation of metabolites and into testing. Especially in this case, where a single key metabolite is the clear signal to test. I only requested one experiment to confirm that the choline observation plays a role in severity of symptoms or as a potential target for therapeutics.

Response: We recognize the effect of choline on disease severity was not directly tested in our study and appreciate the reviewer's concern regarding the correlative nature of this conclusion. That said, we feel our study's primary conclusion and the one most worthy of publication is that the single point mutation leading to FD also perturbs the normal composition and function of the microbiome and metabolome. In addition to evidence in both humans with the disease and a mouse model, we performed an intervention (in mice) that restored microbiome/metabolome function and at least partially reversed the effects on disease progression and severity. This information is novel with potentially significant impact on clinical management of FD.

More importantly, even if an additional experiment failed to support the importance of choline, our primary conclusion regarding the impact of the FD mutation on microbiome/metabolome composition and function would not change; it would merely support alternative hypotheses regarding mechanism. This situation is similar, if not identical, to when fecal microbiota transplantation (FMT) was being developed to treat *Clostridioides difficile* infection. Experimental evidence supported its use in human trials even though the molecular mechanism underlying its efficacy was unknown. In fact, the mechanism responsible for FMT efficacy continues to be debated and explored (PMID: 34665847). Thus, we respectfully disagree that the microbiome field is now at the point of mechanistic understanding. We agree it is an important goal, but this level of understanding in such complex systems is rare and takes years to fully elucidate.

Reviewer #2:

Comment: The concerns from the initial reviews have largely been reasonably addressed in the revisions

Response: We thank the reviewer for their time/effort spent on our manuscript.

Reviewer #3:

Comment: All my concerns have been adequately addressed.

Response: We thank the reviewer for their time/effort spent on our manuscript.